# Development of Procymidone and Difenoconazole Resistance in *Alternaria alternata*, the Causal Agent of Kiwifruit Brown Spot Disease

**DOI:** 10.3390/plants14142245

**Published:** 2025-07-21

**Authors:** Yahui Liu, Manfei Bao, Yanxin Wang, Chuanqing Zhang

**Affiliations:** 1College of Advanced Agricultural Sciences, Zhejiang Agriculture and Forest University, Hangzhou 311300, China; 20070136@zafu.edu.cn (Y.L.);; 2Extension Centre of Agriculture Technology of Hangzhou, Hangzhou 310020, China

**Keywords:** kiwifruit, *Alternaria alternata*, procymidone, difenoconazole, fungicide resistance

## Abstract

Brown spot, caused by *Alternaria alternata*, is the most important leaf fungal disease threatening kiwifruit production in China, and it is typically controlled through the application of fungicides, such as procymidone and difenoconazole. To date, fungicide resistance development has not yet been systematically reported for the pathogen of kiwifruit. A total of 135 single-conidium *A. alternata* isolates were collected from different cities in Zhejiang Province, China. *Alternaria alternata* developed prevailing resistance to procymidone and initial resistance to difenoconazole, with resistance frequencies of 60.7 and 13.3%, respectively. Positive cross-resistance was observed between procymidone and iprodione but not between procymidone and difenoconazole, tebuconazole, prochloraz, pydiflumetofen, pyraclostrobin, or thiophanate-methyl. Moreover, no cross-resistance was observed between difenoconazole and all other tested fungicides, including the two other demethylation inhibitors, tebuconazole and prochloraz. A fitness penalty was not detected in procymidone-resistant (Pro^R^) or difenoconazole-resistant (Dif^R^) isolates. However, double-resistant (Pro^R^ Dif^R^) isolates had a fitness penalty, showing significantly decreased sporulation, germination, and pathogenicity. The P894L single point mutation, caused by the change from CCA to CTA at the 894th codon of *Os1*, was detected in Pro^R^ isolates. Molecular dynamic simulation showed that the P894L mutation significantly decreased the inhibitory activity of procymidone against AaOs1 in *A. alternata*. These results provide insight into the development and characteristics of fungicide resistance, offering guidance for the study and management of kiwifruit diseases.

## 1. Introduction

Kiwifruit (*Actinidia chinensis* Planch.), also known as Chinese gooseberry, is a small fruit that is rich in vitamin C, vitamin K, vitamin E, and potassium [1]. It is native to China and has become popular worldwide [2,3]. China, Italy, New Zealand, Chile, and Greece are the key producers, with a total annual production of 1.8 million tons [4,5]. In 2024, China became the largest global producer of kiwifruit, with a planting area exceeding 200,000 ha and an annual output of more than 3.8 million tons. As an important production and consumption region in China, Zhejiang Province faces significant challenges from foliar diseases [6]. Particularly, brown spot, also called black rot, is caused by *Alternaria alternata* and is the most devastating leaf disease threatening kiwifruit production. *Alternaria alternata* has also been reported to be the causal agent of fruit rot at both preharvest and postharvest stages in kiwifruit [7,8,9]. *Alternaria alternata* is a saprophytic fungal pathogen with a wide host range that includes tomato, apple, pear, orange, strawberry, melon, and tobacco [10,11,12].

Chemical control by fungicides is an important strategy for the management of kiwifruit diseases. Procymidone, a dicarboximide fungicide, classed into Fungicide Resistance Action Committee (FRAC) group 2, inhibits the two-component histidine kinase involved in the osmotic-regulatory signal transduction pathway in fungi [13,14]. It has been applied to control brown spot disease of kiwifruit for more than 10 years. Resistance to group 2 fungicides has been reported in plant pathogens such as *Botrytis cinerea* and *Alternaria* spp. [14,15,16]. Mutations in the osmotic-sensitive-1 (*OS1*) gene are commonly responsible for this resistance [14,17,18]. Difenoconazole (demethylation inhibitor (DMI), FRAC group 3) is also an important agent adopted to manage kiwifruit diseases, including brown spot. Due to the continuous applications of DMI fungicides, some pathogenic fungi have developed resistance, for which *CYP51* mutations are the main molecular mechanisms [14,19,20]. Although significant decreases in control efficacy have been reported by growers and local technicians, the detection and characterization of fungicide resistance have not yet been systematically reported in pathogens of kiwifruit.

This study aimed to (I) assess procymidone and difenoconazole resistance of *A. altenata* on kiwifruit, and (II) evaluate cross-resistance, fitness, and related target mutations. These results provide critical insights for disease control and resistance management on kiwifruit production.

## 2. Results

### 2.1. Procymidone and Difenoconazole Resistance

*Alternaria alternata,* which causes brown spot disease on kiwifruit developed prevailing resistance to procymidone, a typical representative of dicarboximide fungicides, as indicated by a total resistance frequency (F_R_) of 60.7% (Figure 1). Among the four assessed cities, only isolates from Jinhua had an F_R_ value of 33.3%, while isolates from Jiangshan, Shangyu, and Hangzhou had F_R_ values of more than 50.0%. For the tested DMI, difenoconazole, the F_R_ of 13.3% showed that resistance development was still at an initial stage. The F_R_ values of isolates from Jiangshan, Shangyu, Hangzhou, and Jinhua were 8.8, 17.1, 16.7, and 8.3%, respectively (Figure 1).

### 2.2. Cross-Resistance Between Procymidone or Difenoconazole and Other Fungicides

A total of 20 isolates, including both sensitive and resistant isolates, were selected at random to further determine the 50% effective inhibitory concentration (EC_50_) to different fungicides. Procymidone-resistant (Pro^R^) isolates were cultured on potato dextrose agar (PDA) plates amended with 16 μg/mL procymidone (Figure 2). Procymidone-sensitive (Pro^S^) isolates had an EC_50_ of 0.23–0.75 μg/mL, with a mean of 0.41 μg/mL (Appendix A). The five Pro^R^ isolates had a mean EC_50_ of 25.86 μg/mL and a mean resistance factor (RF = EC_50_ of resistant isolate/mean EC_50_ of sensitive isolates) of 62.6. Positive cross-resistance was observed between procymidone and iprodione. The mean EC_50_ to iprodione for the same Pro^S^ and Pro^R^ isolates was 0.52 and 24.68 μg/mL, respectively, with a mean RF of 47.5. All tested isolates had an EC_50_ > 100 μg/mL for thiophanate-methyl, regardless of whether the isolate was sensitive or resistant to procymidone or difenoconazole. Spearman’s rank correlation showed that there was positive cross-resistance between procymidone and iprodione (*ρ* = 0.288, *p* < 0.001). However, no cross-resistance was observed between procymidone and difenoconazole (*ρ* = −0.069, *p* = 0.772), procymidone and tebuconazole (*ρ* = −0.099, *p* = 0.677), procymidone and prochloraz (*ρ* = 0.143, *p* = 0.547), procymidone and pydiflumetofen (*ρ* = −0.295, *p* = 0.206), procymidone and pyraclostrobin (*ρ* = 0.111, *p* = 0.640), and procymidone and thiophanate-methyl (*ρ* = −0.040, *p* = 0.867; Figure 3).

For difenoconazole, the difenoconazole-sensitive (Dif^S^) isolates had EC_50_ values of 0.22–0.76 μg/mL, with a mean of 0.43 μg/mL (Table 1). The difenoconazole-resistant (Dif^R^) isolates had a mean EC_50_ of 15.38 μg/mL and a mean RF of 35.8. No obvious cross-resistance was observed between difenoconazole and iprodione (*ρ* = −0.160, *p* = 0.500), pydiflumetofen (*ρ* =0.048, *p* = 0.841), pyraclostrobin (*ρ* = 0.171, *p* = 0.472), or thiophanate-methyl (*ρ* = 0.296, *p* = 0.205; Figure 4). No positive cross-resistance was observed between difenoconazole and the other two tested DMIs, tebuconazole and prochloraz. The Dif^S^ isolates had EC_50_ values of 0.68–21.1 μg/mL, with a mean of 3.42 μg/mL for tebuconazole. The Dif^R^ isolates had EC_50_ values of 0.95–2.12 μg/mL with a mean of 1.36 μg/mL. There was no significant difference in tebuconazole sensitivity between Dif^S^ and Dif^R^. Similar results were found for prochloraz. Spearman’s rank correlation showed no cross-resistance between difenoconazole and tebuconazole (*ρ* = 0.052, *p* = 0.828) and prochloraz (*ρ* = 0.036, *p* = 0.880; Figure 4).

### 2.3. Fitness of Procymidone- or Difenoconazole-Resistant Isolates

There were significant differences in mycelial growth, sporulation, conidial germination, pathogenicity, and compound fitness index (CFI) among the 17 tested isolates (Appendix A). However, when isolates with two phenotypes (Pro^S^ Dif^S^ and Pro^R^ Dif^S^, Pro^S^ Dif^S^, and Pro^S^ Dif^R^) were compared, there were no significant differences between them (Figure 5 and Figure 6). However, the double-resistant (Pro^R^ Dif^R^) isolates had significantly decreased sporulation, germination, and pathogenicity compared with isolates with one of the other three phenotypes, namely Pro^S^ Dif^S^, Pro^R^ Dif^S^, and Pro^S^ Dif^R^. The CFI of Pro^R^ Dif^R^ also decreased significantly (Figure 7). These results indicate that there was no significant fitness penalty for the Pro^R^ Dif^S^ or Pro^S^ Dif^R^ isolates, but a significant fitness penalty was associated with double resistance (Pro^R^ Dif^R^).

### 2.4. Target Mutations of Os1 and CYP51 in Procymidone- and Difenoconazole-Resistant Isolates

The target coding genes, *Os1* and *CYP51*, for procymidone and difenoconazole, respectively, were amplified using PCR and compared. The P894L single point mutation, caused by the change from CCA to CTA at the 894th codon of *Os1*, was detected in Pro^R^ isolates, including two Pro^R^ Dif^R^ isolates. For *CYP51* and difenoconazole resistance, single nucleotide polymorphisms at the 188th, 192nd, 237th, 307th, 412th, 434th, 448th, and 462nd codons were observed. However, no distinct relationships were observed between these amino acid mutations in *CYP51* and difenoconazole resistance (Table 1).

### 2.5. Binding Affinity Decreased After P893L Mutation of Os1

Four complex systems, wild type (WT) + ADP, WT + procymidone, P894L_ADP, and P894L_procymidone were identified through molecular dynamic simulation. Eleven hydrogen bonds were formed between ADP and N865, N869, A924, D925, T929, T935, L937, and G938 of the AaOs1 WT. In addition, a salt bridge formed between ADP and K872 of WT (Figure 8A). For P894L, seven hydrogen bonds were formed between ADP and N869, A924, D925, T935, G936, or L939, and a π–π conjugate formed with F873 (Figure 8B). Thus, P894L had no effect on the binding affinity of ADP with the Os1 protein. In WT + procymidone and P894L_procymidone, the binding conformation was maintained through hydrophobic interactions and hydrogen bonds (Figure 8C,D). In WT, procymidone bound to the active center of AaOs1 through a hydrophobic interaction with F873, I909, I917, T203, L209, and F965 and hydrogen bonds with Q923 and A924. When P894L mutation occurred, the hydrogen bonds between procymidone and Q923 disappeared, and hydrophobic interactions were weakened. Thus, it was more difficult for procymidone to bind to the active center of AaOs1 in *A*. *alternata*. The P894L mutation significantly decreased the inhibitory activity of procymidone against AaOs1 in *A*. *alternata*.

## 3. Discussion

Procymidone, a FRAC group 2 fungicide, has been applied as the leading chemical to manage major plant diseases caused by *Alternaria* for approximately two decades (http://www.chinapesticide.org.cn/, accessed on 25 December 2024). The present study showed that pathogenic *A. alternata*, which is responsible for brown spot on kiwifruit, had developed widespread resistance to procymidone, with an F_R_ value of 60.7%. Procymidone resistance in *Alternaria* spp. has been reported in China in crops such as garlic, tobacco, broccoli, *Dendrobium officinale*, and *Fritillaria thunbergii* [18,20,21,22,23]. Resistance to group 2 dicarboximides, including procymidone, iprodione, dimethachlon, and chlozolinate, has been reported in other major fungal plant pathogens, such as *B. cinerea*, *Sclerotinia* spp., *Monilinia fructicola*, and *Stemphylium vesicarium* [16,24,25,26,27]. However, resistance of pathogens on kiwifruit, including *A. alternata*, has not been reported or well understood.

Understanding the fitness penalty of the pathogen–fungicide combination will guide resistance management and disease control strategies [19,24,28]. Several studies have reported a fitness penalty for resistance to dicarboximide fungicides. For example, procymidone-resistant *A. alternata* had reduced mycelial growth, pathogenicity, and mycotoxin production [18,19]. In the present study, no significant decrease in growth, sporulation, germination, or pathogenicity was observed for Pro^R^ isolates compared with Pro^S^ isolates. No obvious fitness penalty for resistance to group 2 fungicides has been reported in previous studies [19,28,29,30]. However, a significant fitness penalty was associated with double-resistant (Pro^R^ Dif^R^) isolates in this study.

Different mutations in the *OS1* gene, which encodes a two-component histidine kinase in fungi, are generally to the molecular mechanism of resistance to this fungicide group [19,21,23,26,27]. *Alternaria alternata* on kiwifruit showed positive cross-resistance to iprodione, as previously reported [18,23,27]. Only the P894L mutation in OS1 was observed in Pro^R^ isolates. Molecular dynamic simulation showed that the P894L mutation decreased the inhibitory activity of procymidone against AaOs1. The P894L + S1277L double point mutation has been reported in *A. alternata* isolates from *Fritillaria thunbergii*, a widely cultivated medicinal plant [23].

DMIs (FRAC group 3), such as difenoconazole, are commonly adopted fungicides for the prevention and control of plant diseases caused by *A. alternata.* They target sterol 14α-demethylase (CYP51), the key enzyme in biosterol synthesis [14,19,31]. Resistance to DMIs has been described in many plant pathogens, such as *Aspergillus* spp., *Blumeria graminis*, *Cercospora beticola*, *Colletotrichum* spp., *Fusarium* spp., *Magnaporthe oryzae*, *Monilinia fructicola*, and *Penicillium digitatum* [14,19]. Amino acid mutation in CYP51 is the main resistance mechanism. In our previous study, the G462S substitution of CYP51 was shown to be the main factor for moderate resistance to tebuconazole in *A. alternata* from tobacco, and mechanisms other than CYP51-target mutation might involve isolates with low resistance to tebuconazole [20]. In this study of *A. alternata* from kiwifruit, the initial stage of resistance development to difenoconazole was detected, with an F_R_ value of 13.3%. No cross-resistance was observed between difenoconazole and the other tested fungicides, including two DMIs, tebuconazole and prochloraz. No mutations in CYP51 were found in Dif^R^ isolates, although all tested Dif^R^ isolates were moderately resistant, with a mean RF of 35.8. Our present results indicated the difference in mechanisms of resistance to DMIs for *A. alternata* on kiwifruit with previous reports. No mutations in CYP51 have been previously reported, such as in difenoconazole-resistant mutants of *Sclerotium rolfsii* induced in the laboratory [32]. The I463V point mutation in CYP51A is associated with low difenoconazole resistance in *Colletotrichum truncatum* [33]. The molecular mechanism by which *A. alternata* on kiwifruit resists difenoconazole requires further investigation.

In summary, *A. alternata* on kiwifruit has developed resistance to dicarboximide fungicides associated with the P894L mutation in OS1. In the future, disease management practices of kiwifruit should reduce the use of group 2 dicarboximide fungicides, and strategies such as mixes or rotations with fungicides that have no cross-resistance should be adopted to delay the development of difenoconazole resistance.

## 4. Materials and Methods

### 4.1. Fungicides

The following technical-grade fungicides were used for testing in this study: difenoconazole (a.i. 95.4%, Zhejiang Tianyi Agricultural Chemical Co., Ltd., Hangzhou, China), tebuconazole (a.i. 97%, Zhejiang Yongnong Corporation, Shangyu, China), prochloraz (a.i. 99.5%, Tianfeng Biotech Corporation, Jinhua, China), procymidone (a.i. 98%, Heben Biotech Corporation, Wenzhou, China), iprodione (a.i. 96.1%, Tianfeng Biotech Corporation, Jinhua, China), pydiflumetofen (a.i. 98%, Syngenta, Basel, Switzerland), thiophanate-methyl (a.i. 97%, Zhejiang Welldone Chemistry Company, Hangzhou, China), and pyraclostrobin (a.i. 98%, BASF, Ludwigshafen, Germany). These chemicals were dissolved in methanol or acetone to obtain stock solutions of 10^4^ μg a.i./mL, and then the prepared stock solutions were kept at 4 °C in the dark before further tests.

### 4.2. Origin of Single-Spore Alternaria alternata Isolates

From May to August 2020–2022, kiwifruit leaf samples with typical brown spot disease symptoms were collected from Jiangshan, Shangyu, Hangzhou, and Jinhua. Jiangshan is one of the top 10 main production cities in China. The sampled leaves were placed in plastic bags and stored at 4 °C before isolation. Each leaf was flushed with tap water, and small pieces of tissue (5 mm × 2 mm) were cut from the samples surface-sterilized with 75% ethanol for 30 s and 30% sodium hypochlorite solution for 3 min. Then the tissue was washed three times with sterile distilled water, placed onto PDA (200 g potato, 20 g glucose, 20 g agar, and 1 L distilled water) plates amended with 50 mg/L streptomycin, and incubated at 25 °C in the dark. A total of 135 single-spore *A. alternata* isolates were recovered from 33 orchards. Less than five isolates were collected from each orchard. Procymidone (dicarboximide fungicide, FRAC group 2) and difenoconazole (DMI, FRAC group 3) have been applied on kiwifruit for leaf disease management for approximately 20 and 10 years, respectively. All tested single-spore isolates were identified by pathogenicity on kiwifruit, morphological characteristics (Appendix A), and analysis of the internal transcribed spacer (ITS) sequence [9,20,23]. All isolates were kept on PDA slants at 4 °C in the dark.

### 4.3. Determination of Procymidone and Difenoconazole Resistance

The discriminatory dose method was adopted to identify Pro^S^ and Pro^R^
*A. alternata* isolates, as described in previous studies [23,34]. Pro^S^ isolates had a minimum inhibitory concentration (MIC) of <5 μg/mL for mycelial growth after incubation at 25 °C in the dark for 5 days, and Pro^R^ isolates had an MIC of >5 μg/mL. Similarly, as described in previous studies [20,35], Dif^S^ isolates had an MIC of <5 μg/mL, and Dif^R^ isolates had an MIC of >5 μg/mL. For each concentration, three PDA plates were used per isolate, and the assessments were repeated twice. The resistance frequency was calculated as F_R_/% = (No. of resistant isolates/total no. of tested isolates) × 100.

### 4.4. Determination of Fungicide Sensitivity and Cross-Resistance Analysis

Fungicide sensitivity was determined through the mycelium growth inhibition method. Each isolate was pre-cultured on PDA plates at 25 °C in the dark for 5 days, and 6-mm mycelial plugs were cut from the edge of the colony and placed in the center of PDA plates amended with different concentrations of the tested fungicides (Table 2). Three plates were adopted for each treatment, and the assay was repeated twice. After incubation at 25 °C for 5 days, the diameter of each colony was measured, and the EC_50_ value was calculated for each isolate–fungicide combination by linear regression of the percent inhibition of mycelial growth relative to the control versus the log10 transformation of the fungicide concentrations.

To identify cross-resistance among procymidone, difenoconazole, and other fungicides, 20 isolates, including sensitive and resistant isolates, were randomly selected and assessed. The logEC_50_ of procymidone or difenoconazole was adopted as the x-coordinate, and the logEC_50_ of the other tested fungicides was individually plotted as the y-coordinate to establish the linear regression equations. The Spearman rank correlation was then used to determine cross-resistance. *p* < 0.05 and ρ > 0.6 indicated strong positive cross-resistance between two fungicides [23,36].

### 4.5. Fitness Characterization

Isolates resistant or sensitive to procymidone and difenoconazole were chosen at random to characterize the fitness components, including mycelial growth, sporulation, conidial germination, and pathogenicity as previously described [16,20]. The mycelial growth ability was represented by the mean colony diameter 5 days after incubation on PDA plates at 25 °C in the dark. After incubation for 14 days on PDA plates at 25 °C with continuous light, conidia were recovered with 5 mL of ddH_2_O for each 9-cm PDA plate, and a hemocytometer was used to count the number of conidia, representing sporulation. Then, 100 μL suspensions at a concentration of 1 × 10^5^ conidia per mL were spread onto water agar (WA) plates. After incubation for 18 h at 25 °C in the dark, each plate was evaluated, and the germination rate was calculated. To determine the pathogenicity, healthy kiwifruit leaves were inoculated with *A. alternata* mycelial plugs (6-mm diameter). PDA plugs without mycelia were adopted as the negative control. Five days after incubation at 25 °C with 12 h:12 h (light: dark), the mean lesion size was measured and calculated to represent the pathogenicity. Triplicate measurements were performed for each tested isolate. The compound fitness index (CFI) was calculated as follows: CFI = mycelial growth × conidia production × conidial germination × lesion diameter [20].

### 4.6. Analysis of Mutations in the Coding Gene of Fungicide-Targeted Proteins

Isolates resistant or sensitive to procymidone and difenoconazole were cultured in YEPD medium (3 g yeast extract, 20 g dextrose, and 20 g glucose in 1 L distilled water). After 2 days, genomic DNA was extracted from each isolate using the cetyl trimethyl ammonium bromide (CTAB) method. The gene sequences were PCR amplified using the primers (Appendix A) and procedures previously described for *Os1* and *CYP51* [20,23]. The PCR products were sequenced by Sangon Biotech (Shanghai, China). The coding sequences were translated into amino acid sequences and aligned using the ClustalW multiple alignment through BioEdit (v7.2.6.1, San Diego, CA, USA)

### 4.7. Molecular Docking Analysis

Amino acid sequences of AaOs1(two-component osmosensing histidine kinase) (XP_018380446.1) were downloaded from NCBI (https://www.ncbi.nlm.nih.gov/protein/, accessed on 1 July 2024) and submitted to SWISS-MODEL to construct protein models. To analyze the docking of difenoconazole to AaOs1 before and after P894L substitution, homologous models of the AaOs1 WT and AaOs1-P894L were constructed using the AlphaFold server, an automated protein structure homology modeling server. The structure of difenoconazole was downloaded from the PubChem database and used as the ligand. Autodock Tools 1.5.6 and Gromacs2023.2 were used to perform the molecular dynamic simulation as described by references [20,37,38,39].

### 4.8. Statistical Analysis

The dataset was analyzed using Data Processing System software, version 6.55 (developed by Hangzhou RuiFeng Information Technology Co., Ltd., Hangzhou, China). The least significant difference test was used to identify differences among multiple means, with the significance threshold set at *p* < 0.05.

## 5. Conclusions

Brown spot caused by *A. alternata* on kiwifruit developed prevailing resistance to procymidone while early stage of resistance to difenoconazole. Fitness penalty was detected only in double-resistant (Pro^R^ Dif^R^) isolates. The P894L single point mutation was first reported through significantly decreasing the inhibitory activity of procymidone against AaOs1. No cross-resistance was observed between difenoconazole and tebuconazole or prochloraz. No mutations in CYP51 were found in Dif^R^ isolates, indicating the difference in *A. alternata* on kiwifruit with previous studies. These results provide understanding and guidance for the study and management of fungicide resistance in *Alternaria* and kiwifruit diseases.

## Figures and Tables

**Figure 1 plants-14-02245-f001:**
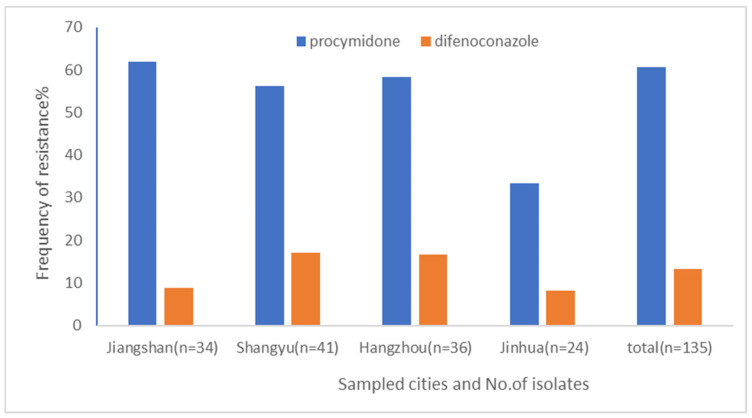
Resistance frequency of *Alternaria alternata* causing brown spot disease on kiwifruit to procymidone and difenoconazole. The number following the sampled city represents the number of isolates obtained in that city. F_R_ (%) = (No. of resistant isolates/total no. of tested isolates) × 100.

**Figure 2 plants-14-02245-f002:**
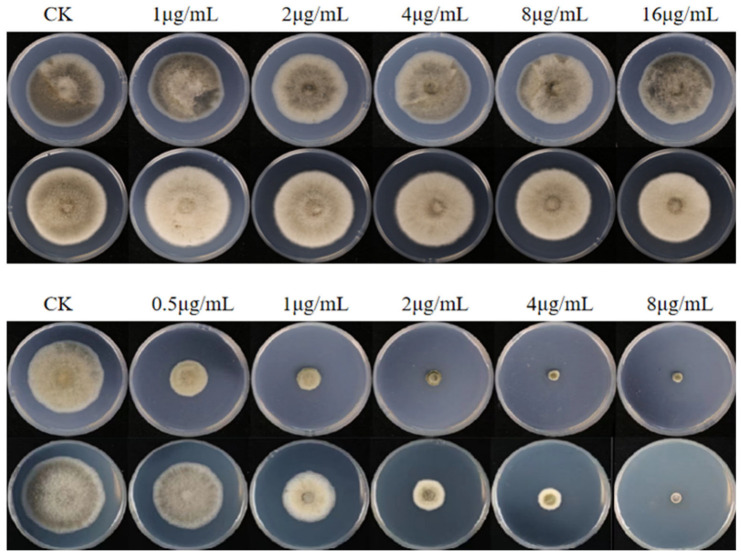
Growth of procymidone-resistant (Pro^R^) (**top**) and procymidone-sensitive (Pro^S^) (**bottom**) isolates with different procymidone concentrations. PDA without fungicide was used as controls (CK).

**Figure 3 plants-14-02245-f003:**
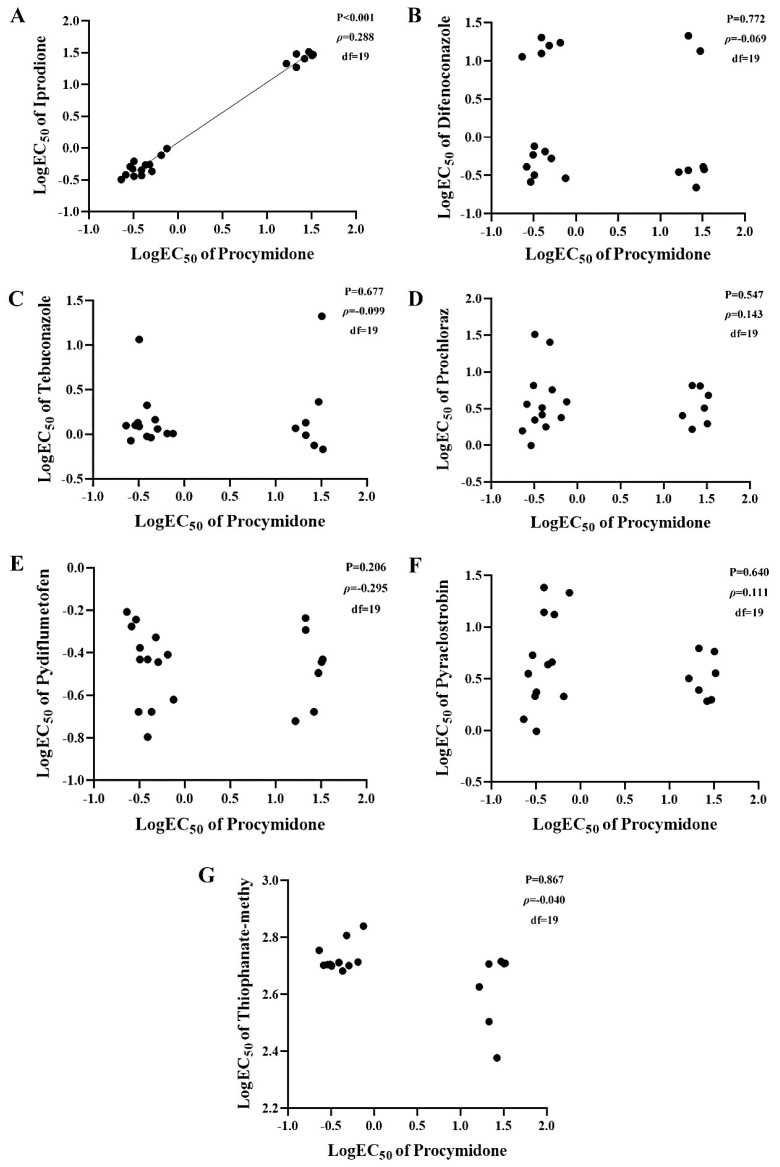
Cross-resistance between procymidone and iprodione (**A**), difenoconazole (**B**), or tebuconazole (**C**) and between procymidone and prochloraz (**D**), pydiflumetofen (**E**), pyraclostrobin (**F**), or thiophanate-methyl (**G**). Spearman’s rank correlation calculations were performed to evaluate the cross-resistance between procymidone and other tested fungicides. *p* < 0.05 and ρ > 0.6 indicated strong positive cross-resistance between two fungicides.

**Figure 4 plants-14-02245-f004:**
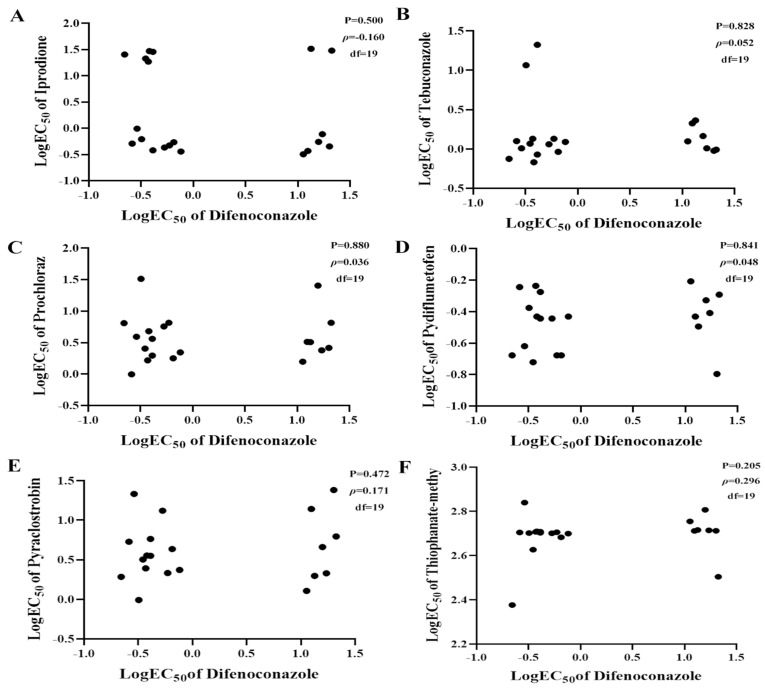
Cross-resistance between difenoconazole and iprodione (**A**) or tebuconazole (**B**) and between procymidone and prochloraz (**C**), pydiflumetofen (**D**), pyraclostrobin (**E**), or thiophanate-methyl (**F**). Spearman’s rank correlation calculations were performed to evaluate the cross-resistance between procymidone and other tested fungicides. *p* < 0.05 and ρ > 0.6 indicated strong positive cross-resistance between two fungicides.

**Figure 5 plants-14-02245-f005:**
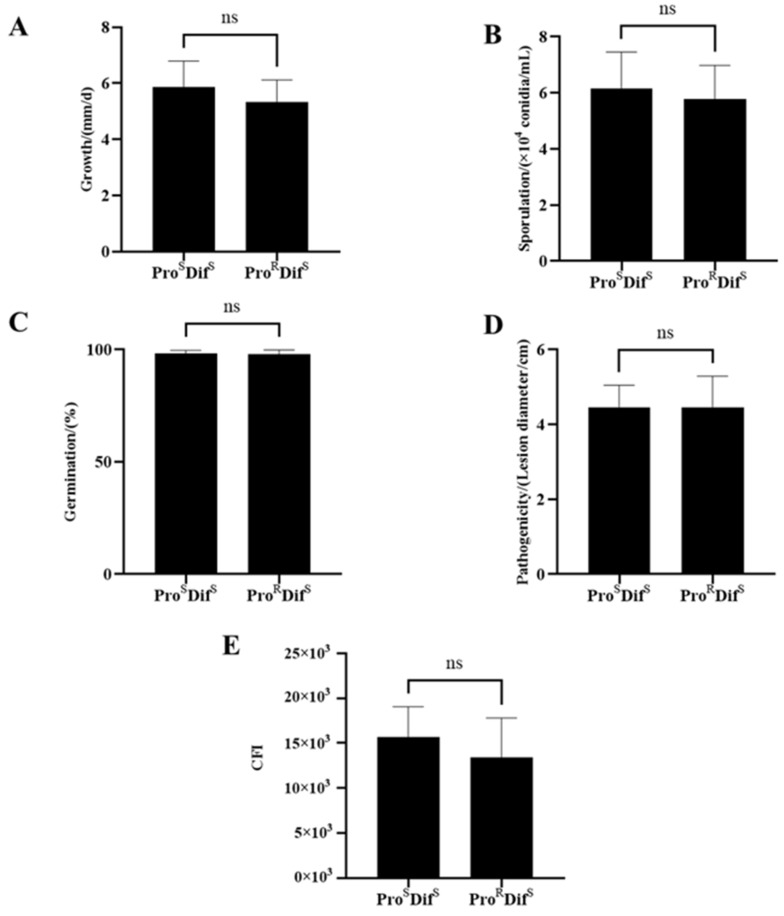
Comparison of mycelial growth (**A**), sporulation (**B**), conidial germination (**C**), pathogenicity (**D**), and CFI (**E**) between procymidone-resistant (Pro^R^) and procymidone-sensitive (Pro^S^) isolates of *Alternaria alternata* from kiwifruit. Dif^S^, difenoconazole-sensitive. Values are shown as the mean ± standard deviation (SD). ns indicates no statistical significance according to Student’s *t*-test.

**Figure 6 plants-14-02245-f006:**
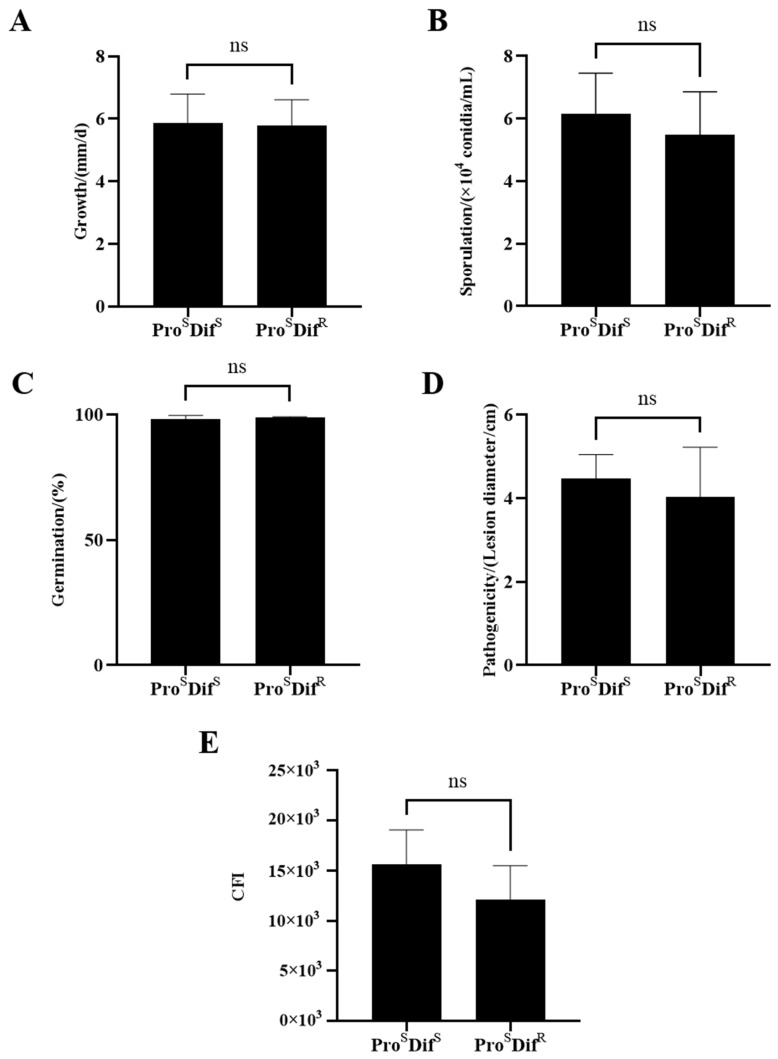
Comparison of mycelial growth (**A**), sporulation (**B**), conidial germination (**C**), pathogenicity (**D**), and CFI (**E**) between difenoconazole-resistant (Dif^R^) and difenoconazole-sensitive (Dif^S^) isolates of *Alternaria alternata* from kiwifruit. Pro^S^, procymidone-sensitive. Values are shown as the mean ± standard deviation (SD). ns indicates no statistical significance according to Student’s *t*-test.

**Figure 7 plants-14-02245-f007:**
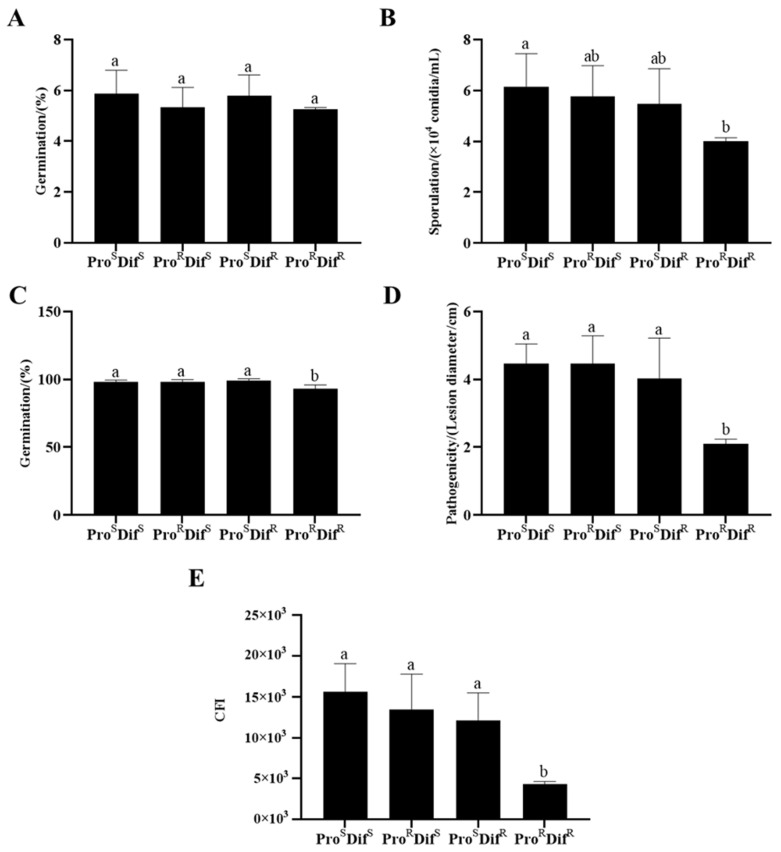
Comparison of mycelial growth (**A**), sporulation (**B**), conidial germination (**C**), pathogenicity (**D**), and CFI (**E**) between the four phenotypes of *Alternaria alternata* isolates from kiwifruit. Pro^S^, procymidone-sensitive; Pro^R^, procymidone-resistant; Dif^R^, difenoconazole-resistant; and Dif^S^, difenoconazole-sensitive. Values are shown as the mean ± standard deviation (SD). Mean values with the same letters are not statistically different (*p* > 0.05) according to the least significant difference test.

**Figure 8 plants-14-02245-f008:**
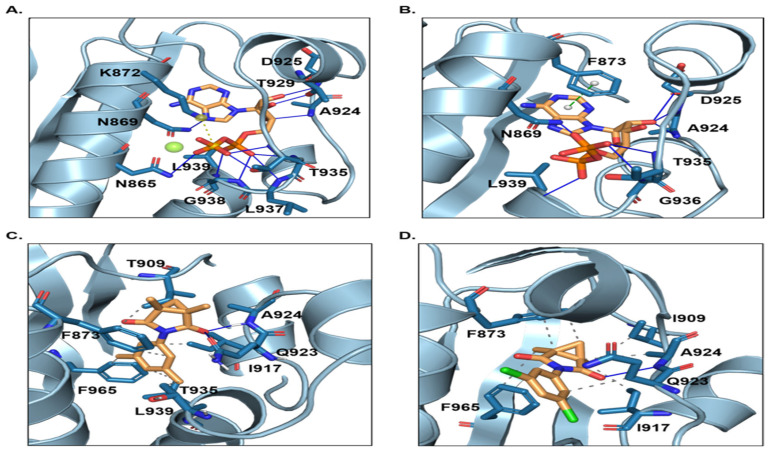
Interaction analysis of OS1. WT (**A**) and P894L (**B**) with ADP; WT (**C**) and P894L (**D**) with procymidone.

**Table 1 plants-14-02245-t001:** Amino acid mutations of Os1 and CYP51 between sensitive and resistant *Alternaria alternata* isolates from kiwifruit.

Isolate	Phenotype	Os1		CYP51							
		894 **	1277	188	192	237	307	412	434	448	462
AK-01	Pro^S^ Dif^S^	P *	S	N	V	S	A	H	E	G	G
AK-08		P	L	N	V	S	G	H	D	G	G
AK-13		P	S	N	V	A	A	Y	E	S	S
AK-22	Pro^R^ Dif^S^	L	S	N	I	S	A	Y	D	G	G
AK-123		L	S	N	V	S	G	H	E	G	S
AK-043		L	S	K	V	A	A	Y	E	S	S
AK-129	Pro^S^ Dif^R^	P	S	K	I	A	G	Y	E	G	G
AK-205		P	L	N	I	S	A	H	D	S	G
AK-024		P	S	N	V	A	G	Y	D	S	S
AK-163	Pro^R^ Dif^R^	L	S	N	V	A	G	Y	E	G	G
AK-037		L	L	K	I	A	G	H	D	G	S

* Proline (P), leucine (L), serine (S), lysine (K), asparagine (N), valine (V), isoleucine (I), alanine (A), glycine (G), tyrosine (Y), histidine (H), glutamic acid (E), and aspartic acid (D).** 894 indicates the 894th amino acid of Os1; the other numbers have a similar meaning.

**Table 2 plants-14-02245-t002:** Fungicide concentrations used for sensitivity determination.

Fungicide	Concentration (µg/mL)
Procymidone	0, 0.25, 0.5, 1, 2, 4, 8, 16
Iprodione	0, 0.25, 0.5, 1, 2, 4, 8, 16
Difenoconazole	0, 0.3125, 0.625, 1.25, 5, 10, 40
Prochloraz	0, 0.15625, 0.3125, 0.625, 1.25, 5, 20
Fenbuconazole	0, 0.15625, 0.3125, 0.625, 1.25, 5, 20
Boscalid	0, 0.3125, 0.625, 1.25, 2.5, 5, 20, 40
Fluxapyroxad	0, 0.3125, 0.625, 1.25, 2.5, 5, 20, 40
Pyraclostrobin	0, 0.3125, 0.625, 1.25, 2.5, 5, 20, 40

## Data Availability

Data will be made available upon reasonable request to the corresponding authors.

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
