# Peer review of "Development of Procymidone and Difenoconazole Resistance in Alternaria alternata, the Causal Agent of Kiwifruit Brown Spot Disease"

_plants, 2025, doi:10.3390/plants14142245_

Round 1

Reviewer 1 Report

Comments and Suggestions for Authors

 The experiments were carried out carefully and the article is very well structured.  The results are excellent and contribute to improvements in the cultivation of kiwi fruit. Few suggestions were inserted in the body of the manuscript.

 Congratulations to the authors for their research. 

Author Response

Reviewer 1:The experiments were carried out carefully and the article is very well structured.  The results are excellent and contribute to improvements in the cultivation of kiwi fruit. Few suggestions were inserted in the body of the manuscript.

 Congratulations to the authors for their research. 

Response: Thanks. Suggestions inserted in the body of the manuscript are all accepted in this revised version.

Reviewer 2 Report

Comments and Suggestions for Authors

I am thankful for the opportunity to review this work. The manuscript has been thoroughly reviewed and meets the standards for publication. The research is original, well-structured, and makes a meaningful contribution to the field. I recommend acceptance for publication with some improvements.

1. You can briefly mention the importance of kiwifruit or the economic impact of brown spot disease to better emphasize the significance of the study in the abstract section. Although resistance hasn't been systematically reported, the study should specify what aspects of resistance are unknown and how this research addresses those gaps. You also have mentioned “significantly improved biomass and nutrient assimilation” in lines 10-12, but it's unclear if the improvement was statistically analysed. Please specify statistical evidence in this section.

2. You have described the problem well in the introduction section; however, the novelty of the current study could be more clearly emphasized. Please state how this study differs from past work on A. alternata resistance in other crops or regions.

3. In the ‘2.1. Procymidone and difenoconazole resistance’ section, there is no statistical analysis provided to confirm whether the resistance differences between cities are significant. In addition, the criteria for classifying isolates as resistant or sensitive are not included in this section and there is no mention of previous resistance monitoring or baseline sensitivity levels for comparison.

4. ‘2.3. The fitness of procymidone- or difenoconazole-resistant isolates’ section doesn't mention variability within groups. Were all ProR DifR isolates equally penalized, or was there a range?

5. ‘2.4. Target mutations of Os1 and CYP51 in procymidone- and difenoconazole-resistant isolates’ section, you can include a correlation heatmap or phylogenetic tree showing whether any of the observed amino acid changes cluster with resistant phenotypes. This study has listed several mutations, e.g., A→G at codon 237, etc., but their possible structural or functional effects on difenoconazole binding are not discussed or modeled. You can use in silico modeling to assess whether any CYP51 substitutions may affect the active site or drug binding affinity.

6. The table 1 presenting Os1 and CYP51 mutations is dense, but it’s tough to interpret. Please highlight mutations specific to resistant isolates in color or bold, and add a column indicating whether each mutation is unique to resistant isolates or found in both resistant and non-resistant isolates.

7.‘3. Discussion’ section, you have noted that no CYP51 mutations were found in DifR isolates, this part is somewhat underdeveloped. You can discuss alternative resistance mechanisms e.g., efflux pumps, overexpression of CYP51, epigenetic regulation, even if speculative, and cite relevant literature. On the other hand, you have drawn comparisons to garlic, tobacco, and broccoli, which is helpful, but it may dilute the kiwifruit-specific focus. You can emphasize the novelty of resistance in kiwifruit and regional significance.

8. In the ‘4.6. Analysis of mutations in the coding gene of fungicide-targeted proteins’ section, please include detailed information on how the mutations were analyzed. Important aspects such as the sequencing platform used (e.g., Sanger or Illumina), reference strains, alignment tools, and software used for mutation calling and annotation are either missing or insufficiently described.

9. In the 4.7. Molecular docking analysis’ section, the preparation of ligands, such as energy minimization, format conversion is not sufficiently described. Please include tools or force fields used. You could mention the PDB ID(s) used for the receptor proteins. This is crucial for validation.

Author Response

Reviewer 2:I am thankful for the opportunity to review this work. The manuscript has been thoroughly reviewed and meets the standards for publication. The research is original, well-structured, and makes a meaningful contribution to the field. I recommend acceptance for publication with some improvements.

  1. You can briefly mention the importance of kiwifruit or the economic impact of brown spot disease to better emphasize the significance of the study in the abstract section. Although resistance hasn't been systematically reported, the study should specify what aspects of resistance are unknown and how this research addresses those gaps. You also have mentioned “significantly improved biomass and nutrient assimilation” in lines 10-12, but it's unclear if the improvement was statistically analysed. Please specify statistical evidence in this section.

Response:  abstract section: Brown spot caused by Alternaria alternata is the most important leaf fungal disease threatening kiwifruit production in China,----These results provide understanding of fungicide resistance development and characteristics, and guidance for the study and management of kiwifruit diseases.

  1. You have described the problem well in the introduction section; however, the novelty of the current study could be more clearly emphasized. Please state how this study differs from past work on  alternataresistance in other crops or regions.

Response: A 5.Conclusions sections was added: Brown spot caused by A. alternata on kiwifruit developed prevailing resistance to procymidone while early stage of resistance to difenoconazole. Fitness penalty was detected only in double-resistant (ProR DifR) isolates. The P894L single point mutation was firstly reported through significantly decreasing the inhibitory activity of procymidone against AaOs1. No cross-resistance was observed between difenoconazole and tebuconazole or prochloraz. No mutations in CYP51 were found in DifR isolates, indicating the difference in A. alternata on kiwifruit with previous studies. These results provide understanding and guidance for the study and management of fungicide resistance in Alternaria and kiwifruit diseases.

  1. In the ‘2.1. Procymidone and difenoconazole resistance’ section, there is no statistical analysis provided to confirm whether the resistance differences between cities are significant. In addition, the criteria for classifying isolates as resistant or sensitive are not included in this section and there is no mention of previous resistance monitoring or baseline sensitivity levels for comparison.

Response: No previous resistance monitoring or baseline sensitivity levels for comparison for Brown spot caused by A. alternata on kiwifruit is available. The discriminatory dose method was adopted to identify sensitive and resistant as described in reference.  4.3. Determination of procymidone and difenoconazole resistance

The discriminatory dose method was adopted to identify ProS and ProR A. alternata isolates, as described in previous studies [23,34]. ProS isolates had a minimum inhibitory concentration (MIC) of  < 5 μg/mL for mycelial growth after incubation at 25°C in the dark for 5 days, and ProR isolates had an MIC of >5 μg/mL. Similarly, as described in previous studies [20,35], DifS isolates had an MIC of <5 μg/mL, and DifR isolates had an MIC of >5 μg/mL. For each concentration, three PDA plates were used per isolate, and the assessments were repeated twice. The resistance frequency was calculated as FR/% = (No. of resistant isolates/total no. of tested isolates) × 100.

  1. ‘2.3. The fitness of procymidone- or difenoconazole-resistant isolates’ section doesn't mention variability within groups. Were all ProR DifR isolates equally penalized, or was there a range?

Response: all ProR DifR isolates equally penalized. Analysis between each isolates was carried out in Table S2.

  1. ‘2.4. Target mutations of Os1 and CYP51 in procymidone- and difenoconazole-resistant isolates’ section, you can include a correlation heatmap or phylogenetic tree showing whether any of the observed amino acid changes cluster with resistant phenotypes. This study has listed several mutations, e.g., A→G at codon 237, etc., but their possible structural or functional effects on difenoconazole binding are not discussed or modeled. You can use in silico modeling to assess whether any CYP51substitutions may affect the active site or drug binding affinity.

Response:single nucleotide polymorphisms at the 188th, 192nd, 237th, 307th, 412nd, 434th, 448th, and 462nd codons were observed between isolates of A. alternata on kiwifruit . However, no any distinct relationships were observed between these amino acid mutations in CYP51 and difenoconazole resistance. That mean this SNP occurred in both sensitive and resistance isolates, no relation with difenoconazole resistance. Therefore, we did not further analyze them.

  1. The table 1 presenting Os1and CYP51 mutations is dense, but it’s tough to interpret. Please highlight mutations specific to resistant isolates in color or bold, and add a column indicating whether each mutation is unique to resistant isolates or found in both resistant and non-resistant isolates.

Response: table 1 was revised.

7.‘3. Discussion’ section, you have noted that no CYP51 mutations were found in DifR isolates, this part is somewhat underdeveloped. You can discuss alternative resistance mechanisms e.g., efflux pumps, overexpression of CYP51, epigenetic regulation, even if speculative, and cite relevant literature. On the other hand, you have drawn comparisons to garlic, tobacco, and broccoli, which is helpful, but it may dilute the kiwifruit-specific focus. You can emphasize the novelty of resistance in kiwifruit and regional significance.

Response: We added: Our present results indicated the difference in mechnisms of resistance to DMIs for A. alternata on kiwifruit with previous reports.

  1. In the ‘4.6. Analysis of mutations in the coding gene of fungicide-targeted proteins’ section, please include detailed information on how the mutations were analyzed. Important aspects such as the sequencing platform used (e.g., Sanger or Illumina), reference strains, alignment tools, and software used for mutation calling and annotation are either missing or insufficiently described.

Response:  the following was added;  The PCR products were sequenced by Sangon Biotech (Shanghai, China). The coding sequences were translated into amino acid sequences and aligned using the ClustalW multiple alignment through BioEdit.

  1. In the 4.7. Molecular docking analysis’ section, the preparation of ligands, such as energy minimization, format conversion is not sufficiently described. Please include tools or force fields used. You could mention the PDB ID(s) used for the receptor proteins. This is crucial for validation.

Response: Ok.

Reviewer 3 Report

Comments and Suggestions for Authors
  1. What is the main question addressed by the research?

The authors of the manuscript investigated the emergence of resistance of kiwifruit brown spot disease to the effects of two widely used fungicides (Procymidone (Pro) and Difenoconazole (Dif)). Procymidone, a dicarboximide fungicide, classed into Fungicide Resistance Action Committee (FRAC) group 2, and difenoconazole, a demethylation inhibitor (DMI), classed into FRAC group 3.

  1. What parts do you consider original or relevant for the field? What specific gap in the field does the paper address?

The detection and characterization of fungicide resistance have not yet been systematically reported in pathogens of kiwifruit. This gap is being addressed by the authors of the manuscript under review.  The authors analyzed the cross-correlation between resistance to procymidone, difenoconazole and other fungicides (iprodione, prochloraz, tebuconazole, pyraclostrobin, pydiflumetofen and thiophanate-methyl).   The studies showed that a significant fitness penalty was associated with double-resistant (ProR, DifR) isolates.  A. alternata on kiwifruit has developed resistance to dicarboximide fungicides associated with the P894L mutation in the OS1 gene. Authors put forward statements: a) the need for reduction in the use of group 2 dicarboximide fungicides; b) strategies such as mixes or rotations with fungicides that have no cross-resistance should be adopted to delay the development of difenoconazole resistance.

  1. What does it add to the subject area compared with other published material?

All of the above are absent in other published materials at least for kiwifruit brown spot disease.

  1. What specific improvements should the authors consider regarding the methodology? What further controls should be considered?

I find it difficult to give advice to scientists working in a field far removed from my area of ​​knowledge and interests.

  1. Please describe how the conclusions are or are not consistent with the evidence and arguments presented. Please also indicate if all main questions posed were addressed and by which specific experiments.

The authors' conclusions about the emergence of resistance of kiwifruit brown spot disease to dicarboximide fungicides FRAC group 2 are supported by their experimental data (Figure 1, Figure 2 etc.). Experimental data on weaker resistance to difenoconazole are also shown. Thus, the authors have proven the emergence of resistance to the fungicide procymidone and proposed strategies for the use of fungicides to prevent the emergence of resistance to the fungicide difenoconazole.

  1. Are the references appropriate?

The 39 references cited in the manuscript are fully relevant to the topic of the study.

  1. Please include any additional comments on the tables and figures and quality of the data.

Figure 2. It is not clear from the caption - (top) and (bottom) refer to the top drawing of the two lines of Petri dishes and to the bottom drawing of the two lines of dishes, or to the top and bottom line of dishes in each of the drawings. But then what do the top and bottom lines of dishes show in each drawing? What does the inscription СК in the upper left corner of the two drawings mean?

Supplementary Materials were not provided either as a separate file or as a working link at the end of the text. Therefore, I cannot say anything about these materials.

Author Response

Reviewer 3:

  1. What is the main question addressed by the research?

The authors of the manuscript investigated the emergence of resistance of kiwifruit brown spot disease to the effects of two widely used fungicides (Procymidone (Pro) and Difenoconazole (Dif)). Procymidone, a dicarboximide fungicide, classed into Fungicide Resistance Action Committee (FRAC) group 2, and difenoconazole, a demethylation inhibitor (DMI), classed into FRAC group 3.

Response: Thanks.

  1. What parts do you consider original or relevant for the field? What specific gap in the field does the paper address?

The detection and characterization of fungicide resistance have not yet been systematically reported in pathogens of kiwifruit. This gap is being addressed by the authors of the manuscript under review.  The authors analyzed the cross-correlation between resistance to procymidone, difenoconazole and other fungicides (iprodione, prochloraz, tebuconazole, pyraclostrobin, pydiflumetofen and thiophanate-methyl).   The studies showed that a significant fitness penalty was associated with double-resistant (ProR, DifR) isolates.  A. alternata on kiwifruit has developed resistance to dicarboximide fungicides associated with the P894L mutation in the OS1 gene. Authors put forward statements: a) the need for reduction in the use of group 2 dicarboximide fungicides; b) strategies such as mixes or rotations with fungicides that have no cross-resistance should be adopted to delay the development of difenoconazole resistance.

Response: Thanks.

  1. What does it add to the subject area compared with other published material?

All of the above are absent in other published materials at least for kiwifruit brown spot disease.

Response: Thanks.

  1. What specific improvements should the authors consider regarding the methodology? What further controls should be considered?

I find it difficult to give advice to scientists working in a field far removed from my area of ​​knowledge and interests.

Response: Thanks.

  1. Please describe how the conclusions are or are not consistent with the evidence and arguments presented. Please also indicate if all main questions posed were addressed and by which specific experiments.

The authors' conclusions about the emergence of resistance of kiwifruit brown spot disease to dicarboximide fungicides FRAC group 2 are supported by their experimental data (Figure 1, Figure 2 etc.). Experimental data on weaker resistance to difenoconazole are also shown. Thus, the authors have proven the emergence of resistance to the fungicide procymidone and proposed strategies for the use of fungicides to prevent the emergence of resistance to the fungicide difenoconazole.

Response: Thanks.

  1. Are the references appropriate?

The 39 references cited in the manuscript are fully relevant to the topic of the study.

Response: Thanks.

  1. Please include any additional comments on the tables and figures and quality of the data.

Figure 2. It is not clear from the caption - (top) and (bottom) refer to the top drawing of the two lines of Petri dishes and to the bottom drawing of the two lines of dishes, or to the top and bottom line of dishes in each of the drawings. But then what do the top and bottom lines of dishes show in each drawing? What does the inscription СК in the upper left corner of the two drawings mean?

Response: Thanks.

Supplementary Materials were not provided either as a separate file or as a working link at the end of the text. Therefore, I cannot say anything about these materials.

Response: Thanks. They are submitted as supplementary files.

Reviewer 4 Report

Comments and Suggestions for Authors

The topic is timely and relevant, and the authors have made a commendable effort. However, I have some comments and suggestions that may help improve the overall quality and clarity of the manuscript.
- Introduction: Before starting with "Brown spot,...", author should connect the two sentences "Zhejiang Province is an important......" and "Brown spot,...". Au
- Objective should rewrite to make it more readable.
- Figure 2: The figures are compressed. Author shouldn't modify the figures. The plate should be totally "round".
- Figure 3, 5, 6, & 7: The figures are compressed. Author shouldn't modify the figures. Correct it.
- 4.1. Fungicides:  Mention their city and country name "Zhejiang Tianyi Agricultural Chemical Co., Ltd., Zhejiang Yongnong Corporation, Heben Biotech Corporation, , Zhejiang Welldone Chemistry Company". Follow this for whole manuscript.
- Alternaria alternata or A. alternata--> Should be italic in whole manuscript.
- 4.2. Origin of single-spore Alternaria alternata isolates: "A total of 135 single-spore A. alternata isolates were recovered from 33 orchards".How you know the isolates are Alternaria alternata. Explain how you idenified "A. alternata isolates"
- Alternaria species, especially those in the Alternaria alternata group, are morphologically and genetically very similar. ITS has low resolution among closely related Alternaria species. ITS alone is not sufficient to definitively identify Alternaria alternata. Auhtors need a multi-locus approach (e.g., ITS + TEF1 + GPD + Alt a1) for accurate species-level identification.
- 4.6. Analysis of mutations in the coding gene of fungicide-targeted proteins: Any specific reason? Isolates resistant or sensitive to procymidone and difenoconazole were cultured in
YEPD medium". What is the purpose of using YEPD medium?
Add a section called "Conclusions."
- Similarity % is too high. The author should reduce below 20%
- I recommend the authors seek assistance from a native English speaker or a language editing service to improve grammar, sentence structure, and academic tone throughout the manuscript.

Comments on the Quality of English Language

I recommend the authors seek assistance from a native English speaker or a language editing service to improve grammar, sentence structure, and academic tone throughout the manuscript.

Author Response

The topic is timely and relevant, and the authors have made a commendable effort. However, I have some comments and suggestions that may help improve the overall quality and clarity of the manuscript.
2.- Introduction: Before starting with "Brown spot,...", author should connect the two sentences "Zhejiang Province is an important......" and "Brown spot,...". Au

Response: Thank you for your suggestion. We have changed the sentence to “As an important production and consumption region in China, Zhejiang Province faces significant challenges from foliar diseases. Particularly, brown spot, also called black rot, is caused by Alternaria alternata and is the most devastating leaf disease threatening kiwifruit production.”
3. - Objective should rewrite to make it more readable.
Response: Thank you for your suggestion. We have changed the objective to “This study aimed to (I) assess procymidone and difenoconazole resistance of A. altenata on kiwifruit, and (II) evaluate cross-resistance, fitness, and related target mutations. These results provide critical insights for disease control and resistance management on kiwifruit production.”

  1. - Figure 2: The figures are compressed. Author shouldn't modify the figures. The plate should be totally "round".

Response: Thank you for your careful suggestion. We have already revised the Figure 2.
-5.Figure 3, 5, 6, & 7: The figures are compressed. Author shouldn't modify the figures. Correct it.

Response: Thank you for your suggestion. We have already revised the Figure 3, 5, 6, & 7.
6. 4.1. Fungicides:  Mention their city and country name "Zhejiang Tianyi Agricultural Chemical Co., Ltd., Zhejiang Yongnong Corporation, Heben Biotech Corporation, , Zhejiang Welldone Chemistry Company". Follow this for whole manuscript.

Response: Thank you for your suggestion. We have modified the fungicides information to” difenoconazole (a.i. 95.4%, Zhejiang Tianyi Agricultural Chemical Co., Ltd., Hangzhou, China ), tebuconazole (a.i. 97%, Zhejiang Yongnong Corporation, Shangyu, China ), prochloraz (a.i. 99.5%, Tianfeng Biotech Corporation, Jinhua, China ), procymidone (a.i. 98%, Heben Biotech Corporation, Wenzhou, China ), iprodione (a.i. 96.1%, Tianfeng Biotech Corporation, Jinhua, China ), pydiflumetofen (a.i. 98%, Syngenta, Basel, Switzerland ), thiophanate-methyl (a.i. 97%, Zhejiang Welldone Chemistry Company, Hangzhou, China ), and pyraclostrobin (a.i. 98·0%, BASF, Ludwigshafen, Germany ).”
7.- Alternaria alternata or A. alternata--> Should be italic in whole manuscript.

Response: Thank you for your careful suggestion. We have carefully checked whole manuscript and and corrected three mistakes to ensure its format is correct.
8.- 4.2. Origin of single-spore Alternaria alternata isolates: "A total of 135 single-spore A. alternata isolates were recovered from 33 orchards". How you know the isolates are Alternaria alternata. Explain how you idenified "A. alternata isolates"

Response:This study did not aim to study the species diversity issue of Alternaria. Most references identify the pathogen of kiwifruit brown spot as Alternaria alternata .In this study,   kiwifruit leaf samples with typical brown spot disease symptoms were collected---All tested single-spore isolates were identified by pathogenicity on kiwifruit, morphological characteristics (Figure S1), and analysis of the internal transcribed spacer (ITS) sequence [9,20,23].

9- Alternaria species, especially those in the Alternaria alternata group, are morphologically and genetically very similar. ITS has low resolution among closely related Alternaria species. ITS alone is not sufficient to definitively identify Alternaria alternata. Auhtors need a multi-locus approach (e.g., ITS + TEF1 + GPD + Alt a1) for accurate species-level identification.

Response: This study did not aim to study the species diversity issue of Alternaria. Most references identify the pathogen of kiwifruit brown spot as Alternaria alternata .In this study,   kiwifruit leaf samples with typical brown spot disease symptoms were collected---All tested single-spore isolates were identified by pathogenicity on kiwifruit, morphological characteristics (Figure S1), and analysis of the internal transcribed spacer (ITS) sequence [9,20,23].
10- 4.6. Analysis of mutations in the coding gene of fungicide-targeted proteins: Any specific reason? Isolates resistant or sensitive to procymidone and difenoconazole were cultured in YEPD medium". What is the purpose of using YEPD medium?

Response: In most common cases, resistance to this two fungicides are caused by mutations in the coding gene of fungicide-targeted proteins. No special  purpose of using YEPD medium, we just use it to culture for extract of DNA.
11.Add a section called "Conclusions."

Response: 5.Conclusions sections was added: Brown spot caused by A. alternata on kiwifruit developed prevailing resistance to procymidone while early stage of resistance to difenoconazole. Fitness penalty was detected only in double-resistant (ProR DifR) isolates. The P894L single point mutation was firstly reported through significantly decreasing the inhibitory activity of procymidone against AaOs1. No cross-resistance was observed between difenoconazole and tebuconazole or prochloraz. No mutations in CYP51 were found in DifR isolates, indicating the difference in A. alternata on kiwifruit with previous studies. These results provide understanding and guidance for the study and management of fungicide resistance in Alternaria and kiwifruit diseases.

Round 2

Reviewer 2 Report

Comments and Suggestions for Authors

The authors have adequately addressed the comments and revised the manuscript.